# Privacy-Preserving Multi-Receiver Certificateless Broadcast Encryption Scheme with De-Duplication

**DOI:** 10.3390/s19153370

**Published:** 2019-07-31

**Authors:** Jianhong Zhang, Peirong Ou

**Affiliations:** 1School of Information Sciences and Technology, North China University of Technology, Beijing 100144, China; 2National Engineering Laboratory for Big Data Collaborative Security Technology, Beijing 100015, China; 3Guangxi Key Laboratory of Cryptography and Information Security, Guilin 541004, China; 4College of Sciences, North China University of Technology, Beijing 100144, China

**Keywords:** data sharing, anonymous broadcast encryption, security proof, secure de-duplication

## Abstract

Nowadays, the widely deployed and high performance Internet of Things (IoT) facilitates the communication between its terminal nodes. To enhance data sharing among terminal devices and ensure the recipients’ privacy protection, a few anonymous multi-recipient broadcast encryption (AMBE) proposals are recently given. Nevertheless, the majority of these AMBE proposals are only proven be securely against adaptively chosen plain-text attack (CPA) or selectively chosen ciphertext attack (CCA). Furthermore, all AMBE proposals are subjected to key escrow issue due to inherent characteristics of the ID-based public cryptography (ID-PKC), and cannot furnish secure de-duplication detection. However, for cloud storage, it is very important for expurgating duplicate copies of the identical message since de-duplication can save the bandwidth of network and storage space. To address the above problems, in the work, we present a privacy-preserving multi-receiver certificateless broadcast encryption scheme with de-duplication (PMCBED) in the cloud-computing setting based on certificateless cryptography and anonymous broadcast encryption. In comparison with the prior AMBE proposals, our scheme has the following three characteristics. First, it can fulfill semantic security notions of data-confidentiality and receiver identity anonymity, whereas the existing proposals only accomplish them by formalizing the weaker security models. Second, it achieves duplication detection of the ciphertext for the identical message encrypted with our broadcast encryption. Finally, it also avoids the key escrow problem of the AMBE schemes.

## 1. Introduction

With development of various Internet of Things (IoT) applications, the communication amongst smart IoT devices has become more and more frequent and convenient. As an important one-to-many communication model, broadcast encryption (BE, for short), which was first formally proposed by Amos Fiat and Moni Naor [1], allows for the broadcaster to deliver the encrypted data to the authorized subset *S* of the receivers that are monitoring the broadcast channel. In addition, only the receivers that belong to the subset *S* can recover the message by their private key, while the other receivers outside of *S* can obtain no information about the delivered data. In general, broadcast encryption is capable of saving more computational complexity and communication overhead than traditional encryption in the peer-to-peer model. Therefore, it has very important applications in communications field [2,3] and IoT [4], etc.

However, IoT devices have some non-negligible vulnerabilities during data sharing and anonymity protection [5,6,7]. At the same time, anonymity is also an important security property in BE schemes, which indicates that any receiver is unable to gain any information of the other receivers’ identity from the ciphertexts. Let us consider an example: a user wants to share some sensitive files with its friends in the cloud; for individual privacy, the user does not want its friends to learn about the others’ identity because they might be the opponent. This problem is very similar to blind carbon copy (BCC) in the email system. To solve this problem, many cryptographers have given many solutions, for instance, Bellare et al.’s public key encryption with key-privacy [8], ciphertext-policy attribute-based encryption with hidden-policy [9], anonymous identity-based encryption [10], anonymous broadcast encryption [11,12,13], and anonymous Certificate-Based Encryption [14,15], where anonymous broadcast encryption is the most efficient method in the multi-user setting. In the cloud environment, anonymity is more important due to its openness. Thus, many applications in keyword search and data retrieval [16,17,18] have considered how to achieve strong anonymity in their schemes. The existing anonymous broadcast encryption schemes are classified into two types, one type is based on public key certificate, and the other type is based on ID-based cryptography. Attribute-based encryption provides scalable encryption while supporting anonymity for users in the same group, that is, with the same attributes [19,20]. They have also been applied widely in cloud computing to support access control for data sharing [21]. However, because of the open problem of revocation in attribute-based encryption, it still suffers from the user revocation in practical application [22,23,24]. Some of the corresponding data could easily be recovered from IoT devices by using forensic techniques [25,26]. Fortunately, Antonis Michalas et al. recently proposed two hybrid encryption schemes [27,28] which can solve the open problem of revocation in attribute-based encryption.

Although cloud storage servers have abundant storage space, the identical data’s different encryption can result in multi-replica; this not only wastes space, but also brings a heavy burden on data maintenance. To save the storage space across multiple users in the cloud storage service, de-duplication is an important candidate technique. However, not all of the public encryption schemes can directly support the de-duplication of the ciphertext since random numbers are introduced in the encrypting process. The convergent encryption and the related security definition have been formalized for addressing the de-duplication of ciphertext [29]. Because random numbers are introduced in the encryption algorithm, it is very difficult that the existing anonymous multi-receiver ID-based broadcast encryption schemes (AMIBE) directly support the de-duplication of ciphertext. To overcome the above de-duplication problem, in this work, we propose a secure Privacy-preserving Multi-receiver Certificateless Broadcast Encryption Scheme with De-duplication. Our construction is characterised as follows: firstly, it is the first anonymous certificateless broadcast encryption scheme with de-duplication; secondly, it is capable of simultaneously achieving confidentiality and anonymity of the receivers’ identities under adaptive CCA security. Thirdly, the key escrow problem does not exist.

## 2. Related Works

In 2006, Barth et al. presented the first public key cryptography-based anonymous BE scheme with chosen-ciphertext security [11]. However, the complexity of decryption linearly grows with the size of the set of the receivers. In 2012, Libert et al. put forth a fully anonymous BE scheme with adaptive chosen-ciphertext security with the random oracle model [30]. Subsequently, Fazio et al. proposed two sublinear ciphertext-size anonymous broadcast encryption schemes in [31] which are proven to be securely against adaptive CPA and adaptive CCA in the standard security model, respectively. In 2007, Delerablee constructed a constant-size ciphertext BE scheme [32]; however, the receivers’ public keys need to be attached in the ciphertext. Until now, the PKC-based anonymous broadcast encryption scheme can achieve constant ciphertext and resist adaptive chosen-ciphertext attack (CCA) in the standard-security model.

ID-based BE (IBBE) is an extension of broadcast encryption in the ID-PKC system [33] in which the user’s public key is replaced with the user’s identity. It simplifies public key management and eliminates the public key certificate. To furnish anonymity protection of the receiver’s identity, the first anonymous multi-receiver identity-based broadcast encryption (AMIBE) scheme [12] was introduced. Nevertheless, their scheme was shown to be insecure by Wang et al. [34] and Chien [35] since it can not achieve anonymity protection of the receiver’s identity, whereafter, Wang et al. also presented a modified proposal to fulfill the anonymity of the receiver’s identity in [34]. Very regretfully, Wang et al.’s modified proposal was pointed to be insecure by Zhang et al. in [36]. In 2018, Tseng et al. presented an improved vision of Fan et al.’s AMIBE by revising receiver anonymity’s security definition in [37] and their scheme was shown to be secure in the random oracle model. In Asia-CCS16, based on the multilinear map, Xu et al. gave an AMIBE scheme which is against anonymity attacks and chosen-plaintext attacks in the standard model [38,39]. However, all multilinear map candidates are broken [40]; thus, their proposal is infeasible in reality. Recently, He et al. proposed an ID-based anonymous BE scheme that can concurrently achieve data indistinguishability and anonymity of the receiver identities under the adaptively chosen ciphertext attacks [41].

ID-based cryptographic protocols cut out complex maintenance of certificates; however, an inherent problem called “*key escrow*” exists. This problem can make the PKG be able to execute any cryptographic operation in the name of users since it knows all users’ private keys. Thus, the problem might result in potential security threats for the ID-based crypto-system. To avoid the key escrow problem, Al-Riyami and Paterson gave a variant of ID-based PKC: certificateless cryptography in [42]. Not only do the advantages of ID-based cryptography remain, but they also prevent the key escrow problem of ID-based PKC. In 2004, Yum et al. presented a general construction construction of certificateless encryption (CLE) [43]. Unfortunately, Yum et al.’s scheme was shown to be insecure by Libert et al. in [44] since it does not satisfy CCA security of CLE. In addition, therewith, Libert et al. put forward a novel construction of CLE achieving CCA security.

Recently, lIslam et al. put forward a pairing-free anonymous multi-receiver certificateless encryption scheme (AMCLE, for short) by combining AMIBE with CLE in [45]. Their scheme can achieve receivers’ anonymity and the ciphertext length is linear with the number of the authorized receivers. When more than one person sends the same data, it will bring a heavy burden to the receivers for data storage. Thus, de-duplication is a wise choice to address the growing demand for storage.

To reconcile de-duplication, Douceur et al. presented a method *convergent encryption* (CE) [46], which is a deterministic symmetric encryption with secret key H(m). If two users Alice and Bob encrypt the same plaintext *m*, they can obtain the same ciphertext $C\;=\;E_{H(m)}\left(m\right)$. Its attractive merit makes it be applied in some commercial system. However, it lacks the detailed security analysis and it is not explicit what its basic security goal precisely is. To solve de-duplication of the identical message which is encrypted under the different secret keys, Bellare et al. put forth a novel notion *Message-Locked Encryption* (MLE) [47]. However, MLE is only capable of providing security of unpredictable data. Recently, Bellare et al. proposed an Interactive message-locked encryption and secure de-duplication [48] which can solve the correlated message’s security problem. Until now, numerous secure de-duplication schemes have been presented for settling data de-duplication in cloud [49,50,51].

## 3. Preliminaries

### 3.1. Bilinear Groups

Throughout the paper, we only consider a Type 2 pairing since our scheme is based on such construction. In the following, we review some concepts of such bilinear group pair.
$G_{1}$ and $G_{2}$ denote two additional groups of the same prime *p*; $G_{T}$ denotes a multiplicative group. In addition, it is deemed to be hard for solving the discrete logarithm problem in group $G_{i},i\in \left\{1,2,T\right\}$.$P_{i}$ denotes the generator of group $G_{i}$, for i∈{1,2}.Let $\varphi :G_{2}\to G_{1}$ be a computable isomorphism map which satisfies $\varphi \left(P_{2}\right)\;=\;P_{1}$; andLet $\hat{e}:G_{1}\times G_{2}\to G_{T}$ denote a computable bilinear map, which meets the following criteria:
Bilinearity: For arbitrary $a,b\in Z_{p}$ and all $Q\in G_{1}$, $F\in G_{2}$, we have $\hat{e}\left(aQ,bF\right)\;=\;\hat{e}{\left(Q,F\right)}^{ab}$;Non-degeneracy: $\hat{e}\left(P_{1},P_{2}\right)\neq 1$.



### 3.2. Security Assumptions

In this subsection, we give several security assumptions [33,52] which are the security foundation to construct the proposed scheme.

**ε-BDH-2 problem [33] in**$\left(G_{1},G_{2}\right)$. Given group elements $a_{1}P_{2},b_{1}P_{2}\in G_{2}$ and $c_{1}P_{1}\in G_{1}$, where $P_{2}\in G_{2}$, $P_{1}\in G_{1}$, and $a_{1},b_{1},c_{1}\in Z_{p}^{*}$; if there exists a PPT-algorithm A which takes $\left(P_{1},P_{2},a_{1}P_{2},b_{1}P_{2},c_{1}P_{1}\right)$ as inputs and outputs, the Type 2 pairing $X\;=\;e{\left(P_{1},P_{2}\right)}^{a_{1}b_{1}c_{1}}\in G_{T}$. A’s advantage is defined as

$$\varepsilon \;=\;Pr[e{\left(P_{1},P_{2}\right)}^{a_{1}b_{1}c_{1}}\leftarrow A\left(P_{1},P_{2},aP_{2},bP_{2},c_{1}P_{1}\right)].$$

We think that ε-bilinear Diffie–Hellman problem in $G_{2}$ and $G_{1}$ holds against A if the algorithm A is not capable of obtaining $\hat{e}{\left(P_{1},P_{2}\right)}^{a_{1}b_{1}c_{1}}$ with a non-negligible probability greater than ε.

**ε-BDDH-2 problem in $\left(G_{1},G_{2}\right)$** [33]. It is hard to distinguish the distributions $D_{1}\;=\left(P_{1},P_{2},a_{1}P_{1},b_{1}P_{1},c_{1}P_{2},e{\left(P_{1},P_{2}\right)}^{a_{1}b_{1}c_{1}}\right)$ and $D_{2}\;=\left(P_{1},P_{2},a_{1}P_{1},b_{1}P_{1},c_{1}P_{2},Z\right)$, where $Z\in G_{T}$ and $a_{1},b_{1},c_{1}\in_{R}Z_{p}$. In general, $D_{1}$ is denoted as the BDDH tuple, and $D_{2}$ is called “random tuple”. For a PPT algorithm B, B’s advantage of breaking the BDDH-2 problem in $\left(G_{2},G_{1}\right)$ is defined as

$$\begin{array}{c} \varepsilon \;=\;|Pr[B\left(P_{1},P_{2},a_{1}P_{1},b_{1}P_{1},c_{1}P_{2},e{\left(P_{1},P_{2}\right)}^{a_{1}b_{1}c_{1}}\right)\;=\;1]\\ -Pr[B\left(P_{1},P_{2},a_{1}P_{1},b_{1}P_{1},c_{1}P_{2},Z\right)\;=\;1]|. \end{array}$$

We think that ε-decisional Bilinear Diffie–Hellman problem in $\left(G_{2},G_{1}\right)$ holds against B if the algorithm B is capable of distinguishing the difference of the above two distributions in a non-negligible probability ε>1/2.

**The Computational Diffie–Hellman problem (CDH) in $G_{1}$**. Let $\left(P_{1},a_{1}P_{1},b_{1}P_{1}\right)\in G_{1}^{3}$ be a random 3-tuple where $a_{1},b_{1}\in Z_{p}$; there does not exist an efficient algorithm A that can calculate $abP_{1}$. *A*’s advantage of breaking the Computational Diffie–Hellman problem in $G_{1}$ is defined as

$$\varepsilon \;=\;Pr[A\left(P_{1},aP_{1},bP_{2}\right)\;=\;abP_{1}].$$

We think that the CDH problem holds against A if the algorithm A is capable of outputting $a_{1}b_{1}P_{1}$ in a non-negligible probability ε.

**The Decisional Diffie–Hellman problem (DDH) in $G_{1}$**. Given a 4-tuple $\left(P_{1},a_{1}P_{1},b_{1}P_{1},W\right)\in G_{1}$ where $a_{1},b_{1}\in Z_{p}$ and $W\in G_{1}$, there does not exist an efficient algorithm A that determines $a_{1}b_{1}P_{1}\;=\;W$. *A*’s advantage of breaking the Decisional Diffie–Hellman problem in $G_{1}$ is defined as

$$\begin{array}{ccc} \varepsilon & \;=\; & |Pr[A\left(P_{1},a_{1}P_{1},b_{1}P_{1},a_{1}b_{1}P_{1}\right)\;=\;1]-\\ & & Pr[A\left(P_{1},a_{1}P_{1},b_{1}P_{1},W\right)\;=\;1]|. \end{array}$$

We think that the DDH problem holds against A if the algorithm A is capable of distinguishing the difference of $a_{1}b_{1}P_{1}$ and *W* in a non-negligible probability ε>1/2.

## 4. Basic System Model and Security Model

### 4.1. System Model

According to the definitions of certificateless encryption and broadcast encryption, we give the basic system model of privacy-preserving multireceiver certificateless broadcast encryption with de-duplication (PMCBED) schemes. The PMCBED scheme mainly borrows the idea in [12,37,38] to achieve privacy protection of receiver identities in the certificateless broadcast encryption scheme and offer the ciphertext de-duplication function. Its framework is showed in Figure 1. It includes four entities: key generation center (KGC), the receivers, the broadcaster and the de-duplicator. Their detailed roles are shown as follows:
KGC: it is a trustworthy entity that is responsible for producing a partial private key of the receiver.the Broadcaster: It is a sender of the message. It first selects a subset of the receivers and calculates the ciphertext of the transmitted message. Afterwards, it sends these ciphertexts to the de-duplicator.The de-duplicator: It is an honest-but-curious entity. It can be acted on by the cloud server. Its goal is to check whether the received ciphertext has its replica existing in the cloud.The Receiver: It is the receiver of the ciphertext, its goal is to decrypt the ciphertext. If and only if it is an authenticated receiver, then it can decrypt the ciphertext.


For a PMCBED scheme, it has eight algorithms: **System-setup**, **Extract partial-private key**, **Set secret-value**, **Set-public-key**, **Set-private-key**, **Encryption**, **Decryption** and **Equality-test**. For each algorithm, its detailed definition is given as follows:
System-setup $\left(1^{\lambda }\right)$. λ is a security parameter, and this algorithm is run by a Key Generation Center (KGC) which takes as input λ, return the public parameters PP and the master secret key msk of KGC. The public parameters PP should be published publicly.Extract partial-private key (msk,ID). In general, this algorithm is run by KGC. It takes as inputs public parameters PP, master key msk and a receiver’s identity ID, and outputs the partial-private key $d_{ID}$ of the receiver.Set secret-value (ID). The algorithm is run by the receiver. It takes as inputs public parameters PP and the identity ID of the receiver, and returns $x_{ID}$ as the receiver’s secret value.Set private-key $\left(x_{ID},d_{ID}\right)$: This algorithm is run by the receiver, it takes as inputs the partial-private key $d_{ID}$ and secret-value $x_{ID}$ of the receiver, and outputs private key $SK_{ID}\;=\;\left(d_{ID},x_{ID}\right)$ of the receiver.Set public-key (ID): The algorithm is used to produce the public key of the receiver. It takes as inputs secret value $x_{ID}$ of the receiver and public parameters PP, and outputs the corresponding public key $Y_{ID}$.Encrypt $\left(m,\left(ID_{1},Y_{ID1}\right),\ldots ,\left(ID_{t},Y_{IDt}\right)\right)$. The broadcaster runs this algorithm by inputting a plaintext m, public parameters PP, a set $S\;=\;\left(ID_{1},Y_{ID1}\right),\ldots ,\left(ID_{t},Y_{IDt}\right)$ of receivers’ identities/public keys, and outputs a ciphertext C=Encrypt(m,params,S).Decrypt (C): The algorithm is run by the receiver. It takes as inputs a ciphertext C, public parameters PP and the private key $SK_{ID}$ of the receiver, returns a recovered message *m* or a symbol *⊥* that indicates decryption error.Equality-test $\left(sk_{TTP},CT,CT^{\prime }\right)$: It is a deterministic algorithm, run by a de-duplicator which is an honest-but-curious entity, it takes public parameter PP, the de-duplicator’s secret key $sk_{TTP}$ and two ciphertexts CT and $CT^{\prime }$ as inputs, and returns 1 if CT and $CT^{\prime }$ are from the identical plaintext, otherwise, returns 0.


### 4.2. Security Models

For a secure public key encryption scheme, it should ensure the confidentiality of the encrypted message, this property is referred to as ciphertext-indistinguishability which can be defined in two security models of chosen-plaintext-attack (CPA) and chosen-ciphertext-attack (CCA) [53]. However, for IND-CPA and IND-CCA, indistinguishability does not hold in a secure de-duplication public key encryption in that it is easily breached by an IND-CPA adversary or an IND-CCA adversary in the game [53]. In the **Challenge phase** of the IND-CPA/CCA security game, the adversary is allowed to select two plaintexts $mt_{0}$ and $mt_{1}$, and then a challenge $C^{*}$ for a plaintext $mt_{b}$ with b∈{0,1} is returned. By invoking the **Equality-test** algorithm, the adversary is able to output the corresponding *b* by computing a ciphertext $\hat{C}$ for plaintext $mt_{b}$ and checking whether $\hat{C}$ matches the challenge ciphertext $C^{*}$. The reason to produce such problem is that, given two ciphertexts, any one can run an **Equality-test** algorithm to check their matching-ability.

To provide IND-CCA security in the public key encryption with de-duplication, a trusted-third party (TTP) is introduced to execute an **Equality-test** algorithm by inputting its private key. Meanwhile, the adversary is not allowed to have access to TTP in the security game. Thus, the **Equality-test** query is not involved in the following security games. In the context of the rest of this paper, we let the de-duplicator act as the TTP.

Inspired by security models of certificateless encryption (CLE) and anonymous BE, the security model of our PMCBED schemes defines two security notations "confidentiality" and " anonymity of the receivers’ identities". For confidentiality, it indicates that an adversary is not capable of obtaining any information of the encrypted message from ciphertext. For anonymity of the receivers’ identities, it indicates that an adversary is not capable of obtaining any identity information of the other receivers from ciphertext.

In the following, we first define the IND-CCA security game for PMCBED. Let $Adv_{I},Adv_{II}$ be Type I and Type II probabilistic polynomial time (PPT) adversaries, respectively. In the following, $Adv_{I}/Adv_{II}$ will make an interactive game with the challenger *C*. 

**Definition** **1.**
*A PMCBED scheme is defined to be secure against adaptive-chosen-ciphertext attack (“IND-CCA security”) if there does not exist a Type I/II of adversaries having a non-ignorable superiority in the following game:*

*Setup: Let λ be a security parameter, C be a Challenger. C invokes a $Setup\;(1^{\lambda })$ algorithm to return public parameters PP and master secret key msk; afterwards, C transmits PP to Adv. If Adv is the Type II adversary $Adv_{II}$, then msk is also sent to Adv. Otherwise, msk is secretly kept by the Challenger and then sends system public parameters PP to adversary Adv who also receives the master secret key msk if it is of Type II. Otherwise, the master secret key msk is kept secret.*

*Phase 1: In this phase, Adv can adaptively make a series of queries:*
−
*Public key query oracle: Upon receiving public key query of the receiver ID, if it is the first query of the receiver, then C invokes **Set public-key** algorithm to produce public key $PK_{ID}$ and return $PK_{ID}$ to Adv. Otherwise, it returns the matching public key in the list.*
−
*Extract partial-private key oracle: On receiving a partial private key query of the receiver ID, C inputs msk to invoke the **Extract partial-private key** algorithm and return $d_{ID}$ if Adv is the Type I $Adv_{I}$; otherwise, the oracle is not required if A is of Type II.*
−
*Extract secret-key oracle: Upon receiving the secret key query of the receiver ID from the adversary Adv, C invokes the **Set secret-value** algorithm to produce secret value $x_{ID}$ and return it to Adv.*
−
*Decrypt oracle: On receiving the decrypting query of (CT,ID) from Adv, C invokes the **Set secret-value** algorithm and **Extract partial-private key** algorithm to obtain private key $SK_{ID}$ of the receiver ID; then, it runs a Decryption $\left(CT,SK_{ID}\right)$ algorithm to recover the corresponding plaintext.*

*Note that when Adv is the Type I $Adv_{I}$, it also needs to query **Public-key-replace oracle** in which the receiver’s public key $Y_{ID}$ is replaced with a new public key $Y_{ID}^{\prime }$ when inputting a receiver’s identity ID and its corresponding public key $Y_{ID}$.*

*Challenge: The adversary Adv submits two distinct equivalent-length messages $m_{0}$ and $m_{1}$ as well as a set of the receivers’ identities/public-keys $S^{*}\;=\;\left(ID_{1}/Y_{1},\ldots ,ID_{k}/Y_{k}\right)$. It is required that Adv cannot query Extract partial-private-key oracle with the identity $ID_{i}\in S^{*}$. The challenger C randomly samples a bit b∈{0,1} to compute the challenge ciphertext $C^{*}\;=\;Encrypt\left(m_{b},PP,S^{*}\right)$ and returns it to adversary A.*

*Phase 2: Adversary Adv can continue to adaptively issue a new sequence of queries as in Phase 1. In addition, $\left(ID^{*}/Y^{*},C^{*}\right)$ is not permitted to issue Decryption query, where $ID^{*}/Y^{*}\in S^{*}$.*

*Meanwhile, in a Type I attack, Adv is not allowed to issue Extract partial-private-key query and Public-key-replace query on identity $ID^{*}$, where $ID^{*}\in S^{*}$.*

*Guess: At last, a guess bit $b^{\prime }\in \left\{0,1\right\}$ is returned by the adversary Adv. Adv wins this game if $b^{\prime }\;=\;b$.*



**Definition** **2.**
*A PMCBED scheme is defined as ANO-CCA security if there does not exist a Type I or II of adversary Adv which has a non-ignorable superiority in the following games:*

***Setup** and **Phase 1**: In the two phases, they are the same as those in the above IND-CCA Game.*

*Challenge: In this phase, Adv produces two challenge sets ${\hat{S}}_{0}$ and ${\hat{S}}_{1}$, where $\left|{\hat{S}}_{1}\left|\;=\;\right|{\hat{S}}_{0}\right|$. In addition, it then submits a message $m^{*}$ and $\left({\hat{S}}_{0},{\hat{S}}_{1}\right)$ to C. In addition, the constraint conditions are as follows: (1) Adv is not permitted to issue Extract partial-private-key queries on $ID^{*}$ when Adv is the Type I adversary $Adv_{I}$,(2) a Adv is not permitted to issue Extract secret-key queries on $ID^{*}$ when Adv is the Type II adversary $Adv_{II}$, where $ID^{*}\in {\hat{S}}_{1}⊕{\hat{S}}_{0}$ and ${\hat{S}}_{1}⊕{\hat{S}}_{0}\;=\;{\hat{S}}_{1}\cup {\hat{S}}_{0}-{\hat{S}}_{0}\cap {\hat{S}}_{1}$. Then, C uniformly samples a bit α∈{0,1} to calculate the ciphertext $C^{*}\;=\;Encrypt\left(PP,{\hat{S}}_{\alpha },m^{*}\right)$ and returns it to Adv.*

*Phase 2. In this phase, Adv adaptively issues a new series of queries as in Phase 1 with the following constraint conditions :(1) Adv is not permitted to issue Extract partial-private-key queries on $ID^{*}$, (2) Public-key-replace queries on $ID^{*}$ are not allowed when Adv is the Type I adversary $Adv_{I}$, (3) Extract secret-key queries on $ID^{*}$ are not allowed when Adv is the Type II adversary $Adv_{II}$, and (4) Adv is not allowed to issue Decryption Query on $\left(ID^{*}/Y^{*};C^{*}\right)$, where $ID^{*}\in {\hat{S}}_{1}⊕{\hat{S}}_{0}$.*

*Guess: At last, a guess bit $\alpha^{\prime }\in \left\{0,1\right\}$ is outputted by Adv. Adv wins this game if $\alpha \;=\;\alpha^{\prime }$.*



## 5. Our Scheme

**Setup:** Let λ be a security parameter, **Setup**
(λ) algorithm takes as input λ, and outputs a bilinear map $e:G_{1}\times G_{2}\to G_{T}$, where $G_{1}$ and $G_{2}$ are two groups satisfying $G_{1}\;=\;<P_{1}>$ and $G_{2}\;=\;<P_{2}>$. In addition, they has the same order *p*. **Note that**
$P_{1}\;=\;\varphi \left(P_{2}\right)$ and $\varphi :G_{2}\to G_{1}$ is an isomorphism. Let $H:{\left\{0,1\right\}}^{*}\to G_{1}$,$H_{1}:G_{T}\leftarrow G_{1}$, $H_{3}:G_{T}\leftarrow Z_{p}$ be three cryptographical hash function. $H_{2}\left(\right)$ and f() are two one-way functions. $H_{0}$ is a random generator of group $G_{2}$. For the KGC, it picks a number $s\in Z_{p}$ at random to calculate its public key $PK_{pub}\;=\;sP_{2}$. Let $TPK\;=\;x_{T}P_{1}$ denote the public key of de-duplicator, $x_{T}\in Z_{p}$ be its private key. (E(·),D(·)) denotes the encryption/decryption algorithm of AES. Finally, the public parameters are Param=
$(P_{1},P_{2},G_{1},G_{2},G_{T},p,\varphi ,TPK,PK_{pub},e,$
$H\left(\right),H_{1},H_{2},H_{3},$
f,
$H_{0},\left(E,D\right)).$
msk=s acts as a master secret key and is kept secretly. 

**Extract partial-private key:** First, in all, a receiver submits its identity ID to the KGC; then, the KGC utilizes its master secret key msk to produce partial-private key $d_{ID}$ of the receiver, where $d_{ID}\;=\;sH\left(ID\right)$. 

**Set secret value:** For a receiver with identifier $ID_{i}$, it uniformly samples a number $xk_{i}\in Z_{p}$ and returns $xk_{i}$ to act as its secret value. 

**Set private-key:** For a receiver with identifier $ID_{i}$, let $d_{ID_{i}}$ be its partial-private key, and $xk_{i}$ be its secret value. Its private key $SK_{ID_{i}}$ is set to be $SK_{ID_{i}}\;=\;\left(xk_{i},d_{ID_{i}}\right)$. 

**Set public-key:** In this algorithm, a receiver with identifier $ID_{i}$ takes an input secret value $xk_{i}$, and outputs its public key $Y_{i}\;=\;xk_{i}P_{1}$. 

**Encrypt:** Given a transmitted message *M* and a group of the receivers with public keys and identifiers ${\left\{ID_{i},Y_{i}\right\}}_{i\;=\;1,2,\ldots ,n}$, a broadcaster computes as follows:
For i=1 to *n*, it calculates $x_{i}\;=\;H_{2}\left(ID_{i}\right)$, and then it produces the polynomial
$$C_{i}\left(x\right)\;=\;\prod_{j\;=\;1,j\neq i}^{n}\frac{x-x_{j}}{x_{i}-x_{j}}\;=\;\sum_{j\;=\;0}^{n-1}b_{i,j}x^{j}\;\mathrm{mod}\;p.$$
Obviously, we find $C_{i}\left(x_{i}\right)\;=\;1$ and $C_{i}\left(x_{j}\right)\;=\;0$ for i≠j.It randomly chooses $k\in Z_{p}$ to compute $C_{1}\;=\;kP_{2}.$Then, it selects $Q\in G_{T}$ and $\tau \in Z_{p}$ to compute $K\;=\;H_{3}\left(Q\right)$ and $C_{3}\;=\;E\left(K,M||\tau \right)$.Next, choose a random number $r_{1}\in Z_{p}$, and then for j∈{1,2,…,n}, it calculates
$$R_{j}\;=\;H_{1}\left(e\left(H\left(ID_{j}\right),k\cdot PK_{pub}\right)\right)+r_{1}Y_{j}.$$In addition, it computes $C_{2}\;=\;e{\left(P_{1},r_{1}P_{2}\right)}^{k}\cdot Q$.In addition, for each t∈{1,2,…,n}, it computes
$$Q_{t}\;=\;\sum_{j\;=\;1}^{n}b_{j,t-1}R_{j}.$$Compute $C_{0}\;=\;\left(f\left(M\right)+f\left(\tau \right)\right)\cdot TPK$ and $C_{-1}\;=\;e{\left(P_{1},P_{2}\right)}^{f(\tau )}$.Finally, the resultant ciphertext is as below:
$$CT\;=\;(C_{-1},C_{0},C_{1},C_{2},C_{3},Q_{1},\ldots ,Q_{n}).$$


**Decrypt:** For a given broadcast-ciphertext $CT\;=\;(C_{-1},C_{0},C_{1},C_{2},C_{3},Q_{1},\ldots ,Q_{n})$, an authorized receiver with identity $ID_{i}$ inputs public parameters Param, system public key $PK_{pub}$ and its private key $SK_{ID_{i}}$ to decrypt broadcast–ciphertext CT by the following steps:
First, it computes $x_{i}\;=\;H\left(ID_{i}\right)$.Then, it calculates
$${\hat{R}}_{i}\;=\;Q_{1}+\sum_{j\;=\;2}^{n}\left(x_{i}^{j-1}Q_{j}\right).$$It computes $W\;=\;xk_{i}^{-1}\cdot \left({\hat{R}}_{i}-H_{1}\left(e\left(sH\left(ID_{i}\right),C_{1}\right)\right)\right)$;In addition, it obtains the decryption key $K^{\prime }\;=\;H_{1}(C_{2}/$
e(W,
$C_{2}$
)).Finally, it obtains the plaintext $M\;=\;D(K^{\prime },C_{3})$ and checks $C_{0}\overset{?}{\;=\;}\left(f\left(M\right)+f\left(\tau \right)\right)TPK$. If it holds, output TRUE.


**Equality-test:** Given two ciphertexts CT and $CT^{\prime }$, where $CT^{\prime }\;=\;\left(C_{-1}^{\prime },C_{0}^{\prime },C_{1}^{\prime },C_{2}^{\prime },C_{3}^{\prime },Q_{1}^{\prime },\ldots ,Q_{n}^{\prime }\right)$ and $CT\;=\;(C_{-1},C_{0},C_{1},C_{2},C_{3},Q_{1},\ldots ,Q_{n})$, the de-duplicator makes use of its private key $x_{T}$ to execute as follows:
(1)$$\begin{array}{ccc} e{\left(C_{0}-C_{0}^{\prime },P_{2}\right)}^{x_{T}^{-1}} & \overset{?}{\;=\;} & C_{-1}/C_{-1}^{\prime }. \end{array}$$

Finally, it returns 1 if the above-mentioned Equation (Equation 1) holds; otherwise, output ⊥.

### Discussion

For the above construction, we can know that, if the receiver’s identity ID is involved in the set of the designated receivers, then this receiver can decrypt the corresponding ciphertext CT since, when this receiver’s identifier satisfies $ID_{i}\in S$, where $S\;=\;\{ID_{1}/PK_{1},\ldots ,ID_{n}/PK_{n}\}$, let $x_{i}\;=\;H\left(ID_{t_{i}}\right),$ we have $C_{j}\left(x_{i}\right)\;=\;0$ for j≠i and

$$\begin{array}{ccc} {\hat{R}}_{i} & \;=\; & Q_{1}+x_{i}Q_{2}+x_{i}^{2}Q_{3}+\ldots +x_{i}^{n-1}Q_{n}\\ & \;=\; & (b_{1,0}R_{1}+b_{2,0}R_{2}+\ldots +b_{n,0}R_{n})+\\ & & x_{i}\left(b_{1,1}R_{1}+b_{2,1}R_{2}+\ldots +b_{n,1}R_{n}\right)+\ldots +\\ & & x_{i}^{n}\left(b_{1,n-1}R_{1}+b_{2,n-1}R_{2}+\ldots +b_{n,n-1}Q_{n}\right)\\ & \;=\; & \left(b_{1,0}+b_{1,1}x_{i}+\ldots +b_{1,n-1}x_{i}^{n-1}\right)R_{1}+\\ & & \left(b_{2,0}+b_{2,1}x_{i}+\ldots +b_{2,n-1}x_{i}^{n-1}\right)R_{2}+\\ & & \ldots +\left(b_{n,0}+b_{n,1}x_{i}+\ldots +b_{n,n-1}x_{i}^{n-1}\right)R_{n}\\ & & \;=\;C_{i}\left(x_{i}\right)R_{i}\;=\;R_{i}. \end{array}$$

Thus, the receiver with identifier $ID_{i}$ is capable of obtaining $r_{1}P_{1}$ by utilizing its partial-private key $d_{ID_{i}}$, namely,

$$r_{1}P_{1}\;=\;{\hat{R}}_{i}-H_{1}\left(e\left(d_{ID_{i}},C_{1}\right)\right).$$

It means that the receiver with identifier $ID_{i}$ is able to decrypt the message by the key $K\;=\;H_{1}\left(C_{2}/e\left(r_{1}P_{1},C_{1}\right)\right)$.

## 6. Security Analysis

In the following theorems, we will show that our aforementioned construction can achieve two security properties: anonymity of the receiver’s identity and confidentiality. 

**Theorem** **1.**
*Let $H,H_{1}$ and $H_{2}$ denote random oracles. If the BDH-2 problem and the DDH problem in $\left(G_{1},G_{2}\right)$ are hard, then our proposed construction can be proven to be secure against the IND-PMCBED-CCA attack of the Type I adversary.*


**Proof.** Suppose there exists a Type *I* of adversary $A_{I}$ in an IND-PMCBED-CCA game. If it can break our construction in a non-negligible probability ϵ, then we are capable of building an algorithm *B* which solves the BDH-2 problem and the DDH problem in $(G_{1},G_{2}$). □

Let $\left(P_{2},aP_{2},bP_{2},cP_{1}\right)$ be a random instance of the BDH-2 problem, where a,b and *c* are unknown random numbers from $Z_{p}$; the target is to compute $e{\left(P_{1},P_{2}\right)}^{abc}$. In addition, let $\left(P_{1},\beta_{1}P_{1},\beta_{2}P_{1},V\right)$ be a random instance of the DDH problem, its target is to determine $V\overset{?}{\;=\;}\beta_{1}\beta_{2}P_{1}$. Therefore, *B* simulates the following security game with the adversary $A_{I}$.

**Setup.** Let $PP\;=\;\{P_{1},P_{2},G_{1},G_{2},e,p,H,H_{1},H_{2},H_{3},\left(E,D\right)\}$ be system parameters; they are built by *B*. In addition, *B* sets $PK\;=\;aP_{1}\;=\;\varphi \left(aP_{2}\right)$ and $TPK\;=\;\beta_{1}P_{1}$. Then, *B* sends public parameters PP to the adversary $A_{I}$. In the following proof, $H_{2}$ acts as a one-way function. $H,H_{1}$ and $H_{3}$ are random oracles.

**Phase 1.** In this phase, $A_{I}$ is capable of adaptively issuing a series of queries.
H-Hash Query: When receiving the H-hash query on $ID_{i}$ from $A_{I}$, *B* answers as below. If a record $ID_{i}$ have appeared in a tuple $\left(ID_{i},Q_{i},\eta_{i},q_{i}\right)$ in the *H*-list which is originally empty, it sends back $Q_{i}$; otherwise, it generates $\eta_{i}\in \left\{0,1\right\}$, and randomly chooses $q_{i}\in Z_{p}$. If $\eta_{i}\;=\;0$, it sets $Q_{i}\;=\;q_{i}P_{1}$, else it sets $Q_{i}\;=\;q_{i}bP_{1}\;=\;q_{i}\cdot \varphi \left(bP_{2}\right)$ and adds $\left(ID_{i},Q_{i},\eta_{i},q_{i}\right)$ in the *H*-list. It returns $Q_{i}$ to $A_{I}$$H_{1}$-Query: On input, an identity $X_{i}$, if $\left(X_{i},T_{i}\right)$ exists in the $H_{1}$-list, then it returns $T_{i}$ to $A_{I}$; otherwise, it picks $T_{i}\in G_{1}$ to return $A_{I}$ and adds $\left(ID_{i},T_{i}\right)$ into the $H_{1}$-list. Note that $H_{1}$-list is originally empty.$H_{3}$-Query: On input, $D_{i}$, if $\left(D_{i},k_{i}\right)$ is in the $H_{3}$-list which being originally empty, it sends back $k_{i}$ to $A_{I}$; otherwise, it picks $k_{i}\in Z_{p}$ to return $A_{I}$ and adds $\left(D_{i},k_{i}\right)$ into the $H_{3}$-list.Public-key query: When $A_{I}$ makes a public key query with $ID_{i}$, if the 3-tuple $\left(ID_{i},Y_{i},xk_{i}\right)$ appears in the PK-list which is initially empty. $Y_{i}$ is returned to $A_{I}$; otherwise, *B* picks $xk_{i}\in Z_{p}$ to set $Y_{i}\;=\;xk_{i}P_{1}$, and adds $\left(ID_{i},Y_{i},xk_{i}\right)$ in the PK-list. Finally, it returns $Y_{i}$ to $A_{I}$.Extract partial-private key Query: Upon receiving a Partial-private key query of the identity $ID_{i}$, if the record $\left(ID_{i},Q_{i},\eta_{i},q_{i}\right)$ had appeared in the *H*-list and $\eta_{i}\;=\;0$, then *B* computes $d_{ID_{i}}\;=\;q_{i}\cdot aP_{1}\;=\;q_{i}\cdot \varphi \left(aP_{2}\right)$. Otherwise, abort it and output *⊥*.Extract secret-value Query: When $A_{i}$ issues a query on an identity $ID_{i}$, if 3-tuple $(ID_{i},Y_{i},$
$xk_{i})$ exists on the PK-list, then $xk_{i}$ is returned to $A_{I}$, otherwise, *B* randomly selects $xk_{i}\in Z_{p}$ to compute $Y_{i}\;=\;xk_{i}P_{1}$ and adds $\left(ID_{i},Y_{i},xk_{i}\right)$ in the PK-list.Public-key-replace Query: When $A_{I}$ makes a public key replace query with $\left(ID_{i},Y_{i}^{\prime }\right)$, the corresponding tuple $\left(ID_{i},Y_{i},xk_{i}\right)$ is replaced into a new tuple $\left(ID_{i},Y_{i}^{\prime },⊥\right)$ in the PK-list.Decryption Queries: On input, a ciphertext CT and an identity $ID_{i}$, where $CT\;=\;(C_{1},C_{2},C_{3},Q_{1}^{\prime },\ldots ,Q_{n}^{\prime })$, *B* first issues a *H*-query with $ID_{i}$ to obtain the tuple $\left(ID_{i},Q_{i},\eta_{i},q_{i}\right)$, if $\eta_{i}\;=\;0$, it sets $d_{ID_{i}}\;=\;q_{i}\cdot P_{1}$ and make a Extract-secret-value query with $ID_{i}$, if $xk_{i}\neq ⊥$ is returned, *B* can make use of $\left(d_{ID_{i}},xk_{i}\right)$ to decrypt CT and respond the Decryption Query. Otherwise, *B* does the following steps:
For j=1 to $q_{H_{3}}${it retrieves $k_{i}$ from $H_{3}$-list and decrypts CT to recover $M||\tau \;=\;D(k_{i},CT)$ with $k_{i}$ to parse it into *M* and τ which can recover τTPK. (**Note that** we assume that the $H_{3}$-query had been made before the adversary issues the decryption-query with CT).if $C_{0}\;=\;f\left(M\right)\cdot TPK+f(\tau )TPK$break;}If $j\leq q_{H_{3}}$, *B* sends back *M* to $A_{I}$. Otherwise, it aborts it.



**Challenge.** In this phase [13], $A_{I}$ submits two equivalent-size plaintext $M_{0}$ and $M_{1}$, as well as a challenge set of identities/public-keys $S^{*}\;=\;(ID_{1}/Y_{1},ID_{2}/Y_{2},\ldots ,ID_{l}/$$PK_{l})$ with the restriction conditions which $A_{I}$ have not issued partial-private-key Oracle with $ID_{i}\in S^{*}$ in phase 1 and each $\eta_{i}\;=\;1$ in the tuple $\left(ID_{i},Q_{i},\eta_{i},q_{i}\right)$ of $H_{1}$-list, where $Y_{i}$ is a public key which corresponds to the identity $ID_{i}$.

Then, *B* computes as follows:
iIt sets $C_{1}^{*}\;=\;cP_{2}$.For j=1 to *l*, it computes $x_{j}^{*}\;=\;H_{2}\left(ID_{j}\right)$.Next, for j=1 to *l*, it constructs the polynomial
$$f_{j}\left(x\right)\;=\;\prod_{i\;=\;1,j\neq i}^{l}\frac{x-x_{i}^{*}}{x_{j}^{*}-x_{i}^{*}}\;=\;\sum_{i\;=\;0}^{l}a_{ji}x^{i}.$$*B* randomly chooses $r_{1}\in Z_{p}$.For j=1 to *l*, it randomly chooses $T_{i}\in G_{1}$ to compute $R_{j}\;=\;T_{j}+r_{1}Y_{j}$.For j=1,2,…,l, *B* computes $Q_{j}\;=\;\sum_{i\;=\;0}^{l}a_{i,j-1}R_{i}$*B* randomly chooses $Q\in G_{T}$ and $\tau \in {\left\{0,1\right\}}^{t}$ to compute $C_{2}^{*}\;=\;e{\left(P_{1},C_{1}^{*}\right)}^{r_{1}}\cdot Q$ and $C_{3}^{*}\;=\;E(K,M_{\beta }\left||\tau \right)$, where $K\;=\;H_{3}\left(Q\right)$, β∈{0,1}.It computes $C_{0}^{*}\;=\;f\left(M_{\beta }\right)TPK+V$ and $C_{-1}^{*}\;=\;e\left(\alpha_{2}P_{1},P_{2}\right)$. Note that in fact $\left(TPK\;=\;\alpha_{1}P_{1},P_{1},V,\alpha_{2}P_{1}\right)$ is also an instance of DDH problem when $\left(P_{1},\alpha_{1}P_{1},\alpha_{2}P,V\right)$ is an instance of DDH problem, since $P_{1}\;=\;\alpha_{1}^{-1}\cdot TPK,V\;=\;\alpha_{2}\cdot TPK$ and $\alpha_{2}P_{1}\;=\;\alpha_{1}^{-1}\alpha_{2}\cdot TPK$.The resultant ciphertext $CT^{*}\;=\;(C_{-1}^{*},C_{0}^{*},C_{1}^{*},C_{2}^{*},$
$C_{3}^{*},Q_{1},\ldots ,Q_{l})$ is returned to $A_{I}$.


**Phase 2.**$A_{I}$ can adaptively make a new series of queries as in Phase 1 with the constraints:
$CT^{*}$ can not be made into Decryption queries.All $ID_{i}\in S^{*}$ is not allowed to issue Extract partial-private-key queries.


**Guess.** Eventually, $A_{I}$ outputs its guess $\beta^{\prime }\in \left\{0,1\right\}$.

When $V\;=\;\beta_{1}\beta_{2}P_{1}$, the challenge ciphertext $CT^{*}$ is a valid one. For the perspective of $A_{I}$, the challenger’s simulation is indistinguishable from the real game. When *V* is a random element of $G_{1}$, the challenge ciphertext has the same distribution as the real ciphertext. Furthermore, we assume that $A_{I}$ must have previously issued $H_{1}$ query with $X_{i}\;=\;e{\left(H\left(ID_{i}\right),PK\right)}^{c}$ Because $C_{1}^{*}\;=\;cP_{1}$, $H\left(ID_{i}\right)\;=\;q_{i}bP_{1}$ and $PK\;=\;aP_{1}$, it means that *B* can compute $e{\left(P_{1},P_{2}\right)}^{abc}\;=\;{\left(X_{i}\right)}^{q_{i}^{-1}}$.

Therefore, it is impossible to have an IND-PMCBED-CCA adversary $A_{I}$ which breaks our PMCBED scheme. □ 

**Theorem** **2.**
*Under the DDH problem in $G_{1}$, our proposed PMCBED scheme is provably secure against the IND-PMCBED-CCA attack of Type II adversary $A_{II}$.*


**Proof.** Assume that there is a Type II of adversary $A_{II}$ in the IND-PMCBED-CCA game. If it breaks our construction, then we are capable of constructing an algorithm *B* to solve the DDH problem. Let $\left(P_{1},aP_{1},bP_{1},Z\right)$ be an instance of DDH problem in group $G_{1}$, where $a,b\in Z_{p}$ are unknown, its goal is to determine $Z\;=\;abP_{1}$. □

**Setup.** Algorithm *B* randomly chooses $\alpha \in Z_{p}$ to compute $PK\;=\;\alpha P_{1}$ and let $TPK\;=\;aP_{1}$. Let PP be public parameters, where $PP\;=\;(P_{1},PK,TPK,e,G_{1},G_{2},P_{2},H,H_{1},H_{2},H_{3},E,D,f)$. Then, it delivers PP and α to the adversary $A_{II}$. $H,H_{1},H_{3}$ are three random oracles which are controlled by *B*.

**Phase 1.**$A_{II}$ can adaptively issue a series of queries.

***H*****-Hash Queries.** Upon receiving a receiver’s identifier $ID_{j}$, *B* first checks that $\left(ID_{j},Q_{j}\right)$ has appeared in the *H*-list which is initially empty; if it is, then $Q_{j}$ is returned. Otherwise, *B* picks $q_{j}\in Z_{p}$ at random to calculate $Q_{j}\;=\;q_{j}P_{1}$ and adds $\left(ID_{j},Q_{j},q_{j}\right)$ in the $H_{1}$-list. Finally, $Q_{j}$ is returned.

**$H_{1}$**-Hash Queries. It is the same as that of Theorem 1.

**$H_{3}$**-Hash Queries. It is the same as that of Theorem 1.

**Public-Key Queries.** Upon receiving an identity $ID_{i}$, if the 3-tuple $\left(ID_{i},Y_{i},xk_{i}\right)$ has existed in the PK-list that was originally empty, then $Y_{i}$ is returned. Otherwise, it produces $\eta_{i}\in \left\{0,1\right\}$ and randomly chooses $a_{i}\in Z_{p}$. If $\eta_{i}\;=\;0$, it sets $Y_{i}\;=\;a_{i}P_{1}$, else it sets $Q_{i}\;=\;a_{i}bP_{1}$ and adds $\left(ID_{i},Y_{i},\eta_{i},a_{i}\right)$ in the PK-list. It returns $Y_{i}$ to $A_{II}$.

**Decryption Query.** Upon receiving $\left(CT,ID_{i}\right)$, if $ID_{i}$ had existed in the PK-list and the corresponding $\eta_{i}\;=\;0$ holds, then *B* decrypts the ciphertext CT by $\left(\alpha \cdot H\left(ID_{i}\right),a_{i}\right)$ and returns the decrypted message *M* to the adversary $A_{II}$. Otherwise, *B* does the following steps:
For j=1 to $q_{H_{3}}${it retrieves $k_{i}$ from $H_{3}$-list and decrypts CT to recover $M\;=\;D(k_{i},CT)$ with $k_{i}$;if $C_{0}\;=\;f\left(M\right)\cdot TPK+f(\tau )TPK$break;}If $j\leq q_{H_{3}}$, *B* sends back *M* to $A_{II}$. If not, it aborts it.


**Challenge Phase.** Let $S^{*}\;=\;(ID_{1}/Y_{1},ID_{2}/Y_{2},\ldots ,ID_{l}$/$Y_{l})$. In this phase, the adversary $A_{II}$ outputs two equivalent length messages $M_{0}$ and $M_{1}$, and a set of identites/public-keys $S^{*}$ with the restriction conditions with each $\eta_{i}$ in the tuple $(ID_{i},Y_{i},\eta_{i},a_{i}),$ where $ID_{i}\in S^{*}$ satisfies $\eta_{i}\;=\;1$.

Then, *B* is computed as below:
It uniformly samples $k\in Z_{p}$ to compute $C_{1}^{*}\;=kP_{2}$ and $C_{0}^{*}\;=\;f\left(M_{\beta }\right)TPK+Z$ as well as $C_{-1}^{*}\;=e\left(bP_{1},P_{2}\right)$. **Note that** we have the relation $\left(D_{0}\;=\;aP_{1},D_{1}\;=\;P_{1}\;=D_{0}^{a^{-1}},D_{2}\;=\;Z\;=\;D_{0}^{b},D_{4}\;=\;bP_{1}\;=\;D_{0}^{a^{-1}b}\right)$ which is the instance of the CDH problem if $Z\;=\;abP_{1}$.For j=1 to *l*, it calculates $x_{i}^{*}\;=\;H_{2}\left(ID_{i}\right)$;Then, for j=1 to *l*, it builds the polynomial
$$f_{j}\left(x\right)\;=\;\prod_{j\neq i}^{l}\frac{x-x_{i}^{*}}{x_{j}^{*}-x_{i}^{*}}\;=\;\sum_{i\;=\;0}^{l}a_{ji}x^{i}.$$For j=1 to *l*, *B* computes
$$R_{j}\;=\;H_{1}\left(e\left(\alpha \cdot H\left(ID_{i}\right),C_{1}^{*}\right)\right)+a_{i}\cdot Z.$$
Note that $r_{1}$ in the original encryption is set as $r_{1}\;=\;a$ but is unknown.For i∈{1,2…,l}, it calculates
$$Q_{i}\;=\;\sum_{j\;=\;1}^{l}a_{j,i-1}R_{j}.$$It randomly selects $Q\in G_{T}$ to compute $K\;=\;H_{3}\left(Q\right)$ and $C_{3}^{*}\;=\;E(K,M_{\beta }\left|\right|x_{Z})$, $x_{Z}$ denotes the *x*-coordination of point *Z*.It computes $C_{2}^{*}\;=\;e{\left(aP_{1},P_{2}\right)}^{k}\cdot Q$.The ciphertext is $CT^{*}\;=\;(C_{-1}^{*},C_{0}^{*},C_{1}^{*},$
$C_{2}^{*},$
$C_{3}^{*},$
$Q_{1},$
$\ldots ,Q_{l}).$


**Phase 2.**$A_{II}$ may issue a new series of queries which is the same as what it did in Phase 1 with the restriction that $CT^{*}$ is not made in the Decryption query.

**Guess.** Finally, $A_{II}$ gives its guess $\beta^{\prime }$. If $\beta \;=\;\beta^{\prime }$, $A_{II}$ wins this game with non-ignorable advantage ε. When $Z\;=\;abP_{1}$, the ciphertext $CT^{*}\;=\;\left(C_{0}^{*},C_{1}^{*},C_{2}^{*},C_{3}^{*},Q_{1},\ldots ,Q_{l}\right)$ is a valid one since

$$\begin{array}{ccc} R_{j} & \;=\; & H_{1}\left(e\left(\alpha \cdot H\left(ID_{i}\right),C_{1}^{*}\right)\right)+a_{i}\cdot Z\\ & \;=\; & H_{1}\left(e\left(\alpha \cdot H\left(ID_{i}\right),C_{1}^{*}\right)\right)+a\left(a_{i}\cdot b\right)P_{1}\\ & \;=\; & H_{1}\left(e\left(\alpha \cdot H\left(ID_{i}\right),C_{1}^{*}\right)\right)+a\cdot Y_{i},\\ C_{2}^{*} & \;=\; & e{\left(aP_{1},P_{2}\right)}^{k}\cdot Q\;=\;e{\left(P_{1},aP_{2}\right)}^{k}\cdot Q,\\ C_{0}^{*} & \;=\; & f\left(M_{\beta }\right)TPK+abP_{1}\;=\;f\left(M_{\beta }\right)TPK+bTPK,\\ C_{-1}^{*} & \;=\; & e{\left(P_{1},P_{2}\right)}^{b}\;=\;e{\left(P_{1},P_{2}\right)}^{\tau }. \end{array}$$

This means that $r_{1}\;=\;a$ and τ=b in the encryption. Thus, if $A_{II}$ breaks our scheme, then *B* is able to solve the DDH problem. □ 

**Theorem** **3.**
*Let hash functions $H,H_{1},H_{3}$ be random oracles. If the decisional bilinear Diffie–Hellman problem (DBDH) is hard, then our construction is able to be proved to be secure against the Type I adversary in the ANON-ID-CCA attack game.*


**Proof.** Let $A_{I}$ be an ANON-ID-CCA adversary. If it breaks the proposed AMCLE scheme in a non-ignorable advantage, then we are capable of building a new algorithm *B* to solve the DBDH problem. □

**Setup.** Firstly, let $PK\;=\;aP_{1}$ act as a master public key, and *B* builds the following parameters $PP\;=\;(G_{1},G_{2},G_{T},H,H_{1},$
$H_{2},H_{3},e,p,f,E,D)$, and delivers PP to the adversary $A_{I}$. $H,H_{1},H_{3}$ are three hash functions that act as random oracles.

**Phase 1.**$A_{I}$ is capable of adaptively making a sequence of security queries which are the same as those in Theorem 1.

**Challenge.** After terminating Phase 1, $A_{I}$ submits a challenge message *M* and two disparate sets of identities/public-keys $S_{0}^{*}\;=\;(ID_{0}^{*}/Y_{0}^{*},ID_{2}/Y_{2},$
…,
$ID_{l}$/$Y_{l})$ and $S_{1}^{*}\;=\;(ID_{1}^{*}/Y_{1}^{*},ID_{2}/$
$Y_{2},$
…,
$ID_{l}$/$Y_{l})$ with the constraint in which $A_{I}$ can not issue **Extract Partial-private-key queries** with $ID_{i}$ for $ID_{i}\in \left\{S_{0}^{*},S_{1}^{*}\right\}$. *B* randomly selects β∈{0,1} to compute as follows:
It sets $C_{1}^{*}\;=\;cP_{2}$.*B* retrieves $\left(ID_{\beta }^{*},Q_{\beta }^{*},\eta_{\beta }^{*},q_{\beta }^{*}\right)$ by issuing a *H*-query on $ID_{\beta }^{*}$, if $\eta_{\beta }^{*}\;=\;0$ holds, then it aborts it and outputs *⊥*; if $\eta_{\beta }^{*}\;=\;1$, then let $Q_{\beta }^{*}\;=\;q_{\beta }^{*}\cdot bP_{1}$ and $X_{\beta }^{*}\;=\;Z^{q_{\beta }^{*}}$. Next, it issues $H_{2}$-queries with $X_{\beta }^{*}$ to obtain $T_{\beta }^{*}$.Compute $x_{\beta }^{*}\;=\;H_{2}\left(ID_{\beta }^{*}\right)$, and for j=2 to *l*, it computes $x_{i}^{*}\;=\;H_{2}\left(ID_{i}\right)$.Next, for j∈{2,3,…,l}, it constructs the polynomial
$$f_{j}\left(x\right)\;=\;\prod_{i\neq j,i\;=\;1}^{l}\frac{1}{x_{j}^{*}-x_{i}^{*}}\cdot \left(x-x_{i}^{*}\right)\;=\;\sum_{i\;=\;0}^{l}a_{ji}x^{i}.$$*B* randomly chooses $r_{1}\in Z_{p}$ and for j∈{2,3,…,l}, it randomly chooses $T_{i}\in G_{1}$ to compute $R_{j}\;=\;T_{j}+r_{1}Y_{j}$; and then it computes $R_{\beta }\;=\;T_{\beta }+r_{1}Y_{\beta }^{*}$.For j∈{β,2,3,…,l}, *B* computes $Q_{j}\;=\;\sum_{i\;=\;0}^{l}a_{i,j-1}R_{i}.$*B* randomly chooses $Q\in G_{T}$ and $\tau \in Z_{p}$ to compute $C_{2}^{*}\;=\;e{\left(P_{1},C_{1}^{*}\right)}^{r_{1}}$ and $C_{3}^{*}\;=\;E(K,M_{\beta }\left|\right|x_{\tau })$, where $K\;=\;H_{3}\left(Q\right)$ and $x_{\tau }$ is the *x*-coordination of point τ·TPK.*B* computes $C_{0}^{*}\;=\;\left(f\left(M_{\beta }\right)+\tau \right)TPK$ and $C_{-1}^{*}\;=\;e{\left(P_{1},P_{2}\right)}^{\tau }$.The ciphertext is $CT^{*}\;=\;(C_{-1}^{*},C_{0}^{*},C_{1}^{*},$
$C_{2}^{*},$
$C_{3}^{*},$
$Q_{1},$
$\ldots ,Q_{l})$ to the adversary $A_{I}$.


**Phase 2.**$A_{I}$ sequentially issues a new series of queries with the following restrictions:
$A_{I}$ can not issue **Extract Partial-Private-Key Queries** with ID, where $ID\in \{ID_{0}^{*},ID_{1}^{*}\}$.$A_{I}$ can not issue **Public-Key Replace** with ID, where $ID\in \{ID_{0}^{*},ID_{1}^{*}\}$.$A_{I}$ can not issue **Decryption Queries** with $\left(ID,CT^{*}\right)$, where $ID\in \{ID_{0}^{*},ID_{1}^{*}\}$.


**Guess.** Finally, $A_{I}$ outputs its guess $\beta^{\prime }$. *B* outputs 1 when $\beta \;=\;\beta^{\prime }$, it means that $Z\;=\;e{\left(P_{1},P_{2}\right)}^{abc}$; if $\beta \neq \beta^{\prime }$, outputs 0, it means $Z\neq e{\left(P_{1},P_{2}\right)}^{abc}$.

**Analysis:** In the above game, the simulation is indistinguishable from the scheme. If $Z\;=\;e{\left(P_{1},P_{2}\right)}^{abc}$, then we let $k^{*}\;=\;c$. All this time, $CT^{*}$ has the same distribution as the ciphertext in the real game; If *Z* is a random element in $G_{T}$, then the ciphtertext has the uniform distribution in the ciphertext space since $C_{3}^{*}\;=\;E(K,x_{\tau }\left|\right|M_{\beta })$, where $K\;=\;H_{3}\left(Q\right)$ is a random element. Thus, in the adversary $A_{I}$’s view, $M_{\beta }$ is independent, and it cannot provide any information to $A_{I}$. □ 

**Theorem** **4.**
*Let hash functions $H,H_{1}$ and $H_{3}$ be a random oracle. If the DDH assumption in groups $\left(G_{1},G_{2}\right)$ is difficult, then our construction is proven to be secure against the Type II of adversary $A_{II}$ in the ANON-ID-CCA attack game.*


**Proof.** Let $A_{II}$ be an adversary. If it breaks our construction, then we are capable of constructing a novel algorithm *B* which solves the DDH problem. Let $\left(P_{1},aP_{1},bP_{1},Z\right)$ be a random instance of DDH problem in groups $\left(G_{1},G_{2}\right)$, where $a,b\in Z_{p}$ are unknown, its goal is to determine $Z\;=\;abP_{1}$. □

**Setup.** Algorithm *B* randomly chooses $\alpha \in Z_{p}$ to set $PK\;=\;\alpha P_{1}$. Let $PP\;=(P_{1},P_{2},e,p,PK,f,G_{1},G_{2},H,H_{1},H_{2},H_{3},\left(E,D\right))$ denote public parameters that are built by *B*. Then, it delivers PP and α to the adversary $A_{II}$. Here $H,H_{1},H_{3}$ are three random oracles that are controlled by *B*.

**Phase 1.**$A_{II}$ is capable of issuing a series of the same queries as those of Theorem 2.

**Challenge.**$A_{II}$ outputs a challenge plaintext $M^{*}$ and two different sets $S_{0}^{*}$ and $S_{1}^{*}$ of identities/public-keys, where $S_{0}^{*}\;=\;(ID_{0}^{*}/Y_{0}^{*},ID_{2}/Y_{2},$…,$ID_{l}$/$Y_{l})$ and $S_{1}^{*}\;=\;(ID_{1}^{*}/$
$Y_{1}^{*},$
$ID_{2}/$$Y_{2},$
…,
$ID_{l}$/$Y_{l})$. In addition, the following constraints need to be satisfied: $A_{II}$ cannot issue **Extract partial-private-key queries** on $ID_{i}$ in **Phase 1**, where $ID_{i}\in \left\{ID_{0}^{*},ID_{1}^{*}\right\}$. In addition, then *B* randomly selects β∈{0,1} to compute as below:
First, it makes a **Public-key Query** on $ID_{\beta }^{*}$ to obtain $\left(ID_{\beta }^{*},Y_{\beta }^{*},\eta_{\beta }^{*},a_{\beta }^{*}\right)$. If $\eta_{\beta }^{*}\;=\;0$, output *⊥* and abort it. If $\eta_{\beta }^{*}\;=\;1$, it means that $Y_{\beta }^{*}\;=\;a_{\beta }^{*}\cdot bP_{1}$.For j∈{η,2,3,…,l}, it calculates $x_{i}^{*}\;=\;H_{2}\left(ID_{i}\right)$;Then, for j∈{β,2,3,…,l}, it builds the polynomial
$$f_{j}\left(x\right)\;=\;\prod_{i\neq i}^{l}\frac{x-x_{i}^{*}}{x_{j}^{*}-x_{i}^{*}}\;=\;\sum_{i\;=\;0}^{l}a_{ji}x^{i}.$$For j∈{β,2,3,…,l}, *B* issues **Public-key Queries** with $ID_{j}$ to obtain $\left(ID_{j},Y_{j},\eta_{j},a_{j}\right)$. If $\eta_{\beta }^{*}\;=\;0$, *B* computes
$$R_{j}\;=\;H_{1}\left(e\left(\alpha \cdot H\left(ID_{j}\right),C_{1}^{*}\right)\right)+a_{j}\cdot aP_{1}.$$
If $\eta_{\beta }^{*}\;=\;1$, it computes $R_{j}\;=\;H_{1}\left(e\left(\alpha \cdot H\left(ID_{j}\right),C_{1}^{*}\right)\right)+a_{j}\cdot Z.$For j∈{β,2,3,…,l}, it computes $Q_{i}\;=\;\sum_{j\;=\;1}^{l}a_{j,i-1}R_{j}.$It randomly selects $Q\in G_{T}$ and $\tau \in Z_{p}$ to compute $K\;=\;H_{3}\left(Q\right)$ and $C_{3}^{*}\;=\;E(K,x_{\tau }\left|\right|M_{\beta })$.It randomly chooses $k\in Z_{p}$ to compute $C_{1}^{*}\;=\;kP_{2}$ and $C_{0}^{*}\;=\;f\left(M_{\beta }\right)+\tau \cdot TPK$ as well as $C_{-1}^{*}\;=\;e{\left(P_{1},P_{2}\right)}^{\tau }$.It computes $C_{2}^{*}\;=\;e{\left(aP_{1},P_{2}\right)}^{k}\cdot Q$.The resultant ciphertext is $CT^{*}\;=\;(C_{-1}^{*},C_{0}^{*},$
$C_{1}^{*},$
$C_{2}^{*},$
$C_{3}^{*},$
$Q_{1},$
…,
$Q_{l}).$


**Phase 2.**$A_{II}$ can still adaptively issue the queries with the following constraints.
$A_{II}$ is not capable of issuing **Public-key Query** with ID, where $ID\in \{ID_{0}^{*},ID_{1}^{*}\}$.$A_{II}$ is not capable of issuing **Decryption Query** with $\left(CT^{*},ID\right)$, where $ID\in \{ID_{0}^{*},ID_{1}^{*}\}$.


**Guess.** Finally, $A_{II}$ returns its guess bit $\beta^{\prime }$. *B* outputs 1 if $\beta \;=\;\beta^{\prime }$; it means that $Z\;=\;abP_{1}$; otherwise, outputs 0 meaning $Z\neq abP_{1}$.

**Analysis:** In the above game, the simulation is indistinguishable from the scheme. When $Z\;=\;abP_{1}$, assume $r_{1}\;=\;a$. The challenge ciphertext has the same distribution as that in the real game, in addition to when *Z* is a random element of $G_{1}$, $C_{2}^{*}$ and $C_{3}^{*}$ in the ciphtertext has the form $C_{2}^{*}\;=\;e{\left(aP_{1},P_{2}\right)}^{k}\cdot Q$ and $C_{3}^{*}\;=\;E(K,x_{\tau }\left|\right|M_{\beta })$, where $K\;=\;H_{3}\left(Q\right)$ and *Q* are uniform and random. Thus, from the adversary $A_{II}$’s view, $M_{\beta }$ is independent; it provides no information to $A_{II}$. □

## 7. Performance Analysis

To evaluate the efficiency of the proposed scheme, we give the corresponding computational cost of the main algorithm by comparing with the Hung et al. scheme [37] and Islam et al. scheme [45]. For convenience, we define the following notations. Let $T_{p},T_{m},T_{e}$ and $T_{h}$ denote the time of executing a pairing operation, a scalar multiplication operation and an exponentiation operation as well as a map-to-point hash function, respectively. The computation cost of the main algorithms for the three schemes are shown in Table 1.

From Table 1, we find that our proposed scheme has more computational costs than the other two schemes. However, our proposed scheme has better security and functionality.

## 8. Conclusions

The users are increasingly concerned about anonymity. To protect the identity anonymity of the receiver, we construct a privacy-preserving Multi-receiver Certificateless Broadcast Encryption Scheme with De-duplication scheme in this work. It can not only simultaneously achieve confidentiality and the receiver’s identity anonymity, but also achieve duplicate detection to determine whether two different ciphertexts are from the identical message. Thus, our proposal can efficiently reduce the cloud server’s storage burden. It is very significant for cloud storage. Nevertheless, the ciphertext size is linear to the number of the receivers. A very important challenge will be how to construct a PMCBED scheme with constant-size ciphertext.

## Figures and Tables

**Figure 1 sensors-19-03370-f001:**
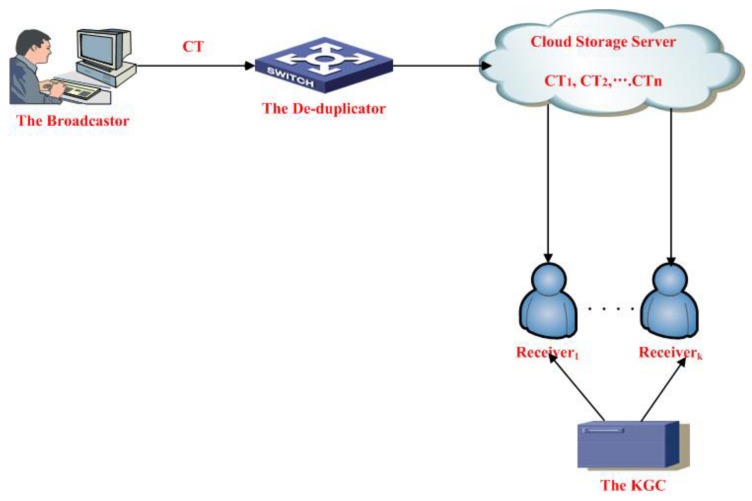
The system model of the PMCBED scheme.

**Table 1 sensors-19-03370-t001:** Comparison of computation costs in the three schemes.

	Islam et al. Scheme [45]	Hung et al. Scheme [37]	Our Scheme
Computational cost of encryption for *n* receivers	$\left(2n+1\right)T_{p}+\left(n^{2}+n\right)T_{m}$	$nT_{p}+nT_{e}+\left(n+1\right)T_{m}+nT_{h}$	$\left(n+1\right)T_{p}+\left(n+2\right)T_{m}+2T_{e}+nT_{h}$
Complexity of encryption	$O(n^{2})$	O(n)	$O(n^{2})$
Computational cost of decryption for each receiver	$T_{m}+nT_{h}$	$T_{p}+1T_{M}$	$3T_{p}+\left(n+3\right)T_{M}+T_{e}$
Complexity of decryption	O(n)	O(1)	O(n)
De-duplication	No	No	Yes
Security	selective-CCA security	selective-CCA security	CCA-security

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
