# Peer review of "Privacy-Preserving Multi-Receiver Certificateless Broadcast Encryption Scheme with De-Duplication"

_sensors, 2019, doi:10.3390/s19153370_

Round 1

Reviewer 1 Report

This paper focuses on the very important problem of deduplication of data in cloud storage. Authors present a privacy-preserving multi-receiver certificateless broadcast encryption scheme that is applied to a cloud environment. The proposed scheme has three key characteristics:

- Is semantically secure and achieves data-confidentiality while at the same time can protect the identity of the receiver.
- Can successfully identify duplicate ciphertexts.
- Does not face the key escrow problems that other similar schemes suffer from.

While authors have thoroughly described their scheme as well as the basic cryptographic primitives, they have not mentioned anything about a possible system model under which this scheme could be applied. To my eyes, a trusted cloud service provider that will be based on the principles of Trusted Computing [1] - such as the one described in [2] seems like the best fit. In addition to that, authors have missed important works from their related work part. For example, authors mention that "However, because of the open problem of revocation in attribute-based encryption, it still suffers from the user revocation in practical application [24–26].". It looks like authors are missing important references works in the area. More precisely, lately a workaround has been published [3, 4] where it separates the revocation mechanism from the actual ABE scheme. The authors should not just talk about the problem without known solutions.

While the security analysis is quite formal and rigorous there is absolutely no implementation or performance testing. Therefore, it is impossible to quantify the efficiency of the proposed scheme. However, since efficiency is the main selling point of the underlying scheme, it makes more sense for authors to think about how to quantify the performance of their scheme. Moreover, without a real-world setup, it is hard for readers to believe that the assumption made in the paper is practical. I think a real-world performance evaluation/testing or at least a lab-based performance evaluation must be added.

Finally, while I appreciated the security analysis and the definition of the underlying security models the paper has many typos and grammatical mistakes. Therefore, the authors need to properly revisit the manuscript (on top of addressing the previous comments).

References
[1] Nuno Santos, Krishna P. Gummadi and Rodrigo Rodrigues. "Towards trusted cloud computing". In Proceedings of the 2009 conference on Hot topics in cloud computing (HotCloud'09). USENIX Association, Berkeley, CA, USA, 2009.

[2] Nuno Santos, Rodrigo Rodrigues, Krishna P. Gummadi and Stefan Saroiu. "Policy-sealed data: a new abstraction for building trusted cloud services". Proceeding Security'12 Proceedings of the 21st USENIX conference on Security symposium Pages 10-10 Bellevue, WA — August 08 - 10, 2012

[3] Alexandros Bakas and Antonis Michalas. “Modern Family: A Revocable Hybrid Encryption Scheme Based on Attribute-Based Encryption, Symmetric Searchable Encryption and SGX”. In Proceedings of the 15th EAI International Conference on Security and Privacy in Communication Networks (SecureComm’19). Orlando, United States, October 23 – 25, 2019.

[4] Antonis Michalas. “The Lord of the Shares: Combining Attribute-Based Encryption and Searchable Encryption for Flexible Data Sharing”. In Proceedings of the 34th ACM/SIGAPP Symposium On Applied Computing (SAC). Limassol, Cyprus, April 08 – 12, 2019.

Author Response

We would like to thank the editor and the anonymous reviewers for providing us with the chance to revise our work. In the following, we address each comment in detail. And the revised parts are marked in red font.

Review1

1.       in abstract : “economize”………..>”save”

2.       in abstract: “deal with”……………….>”address”

- Is semantically secure and achieves data-confidentiality while at the same time can protect the identity of the receiver.

Answer: the proposed scheme is semantically secure, and can achieve data-confidentiality. In theorem 1 and theorem 2, we give the corresponding security proof.  It can realize anonymity of the receiver since our scheme is based on anonymous broadcast encryption

- Can successfully identify duplicate ciphertexts.

Answer: it can achieve ciphertext Deduplication

- Does not face the key escrow problems that other similar schemes suffer from.

Answer: in this paper, we avoid key escrow by using certificateless cryptography.

While authors have thoroughly described their scheme as well as the basic cryptographic primitives, they have not mentioned anything about a possible system model under which this scheme could be applied. To my eyes, a trusted cloud service provider that will be based on the principles of Trusted Computing [1] - such as the one described in [2] seems like the best fit. In addition to that, authors have missed important works from their related work part. For example, authors mention that "However, because of the open problem of revocation in attribute-based encryption, it still suffers from the user revocation in practical application [24–26].". It looks like authors are missing important references works in the area. More precisely, lately a workaround has been published [3, 4] where it separates the revocation mechanism from the actual ABE scheme. The authors should not just talk about the problem without known solutions.

Answer: Very thank the reviewer’s suggestions. Due to systematic understanding on attribute-based encryption, it results in the missing knowledge. According to the reviewer’s suggestions, we add the corresponding content.  And we also add related work.

While the security analysis is quite formal and rigorous there is absolutely no implementation or performance testing. Therefore, it is impossible to quantify the efficiency of the proposed scheme. However, since efficiency is the main selling point of the underlying scheme, it makes more sense for authors to think about how to quantify the performance of their scheme. Moreover, without a real-world setup, it is hard for readers to believe that the assumption made in the paper is practical. I think a real-world performance evaluation/testing or at least a lab-based performance evaluation must be added.

Answer:  Thank your suggestions, I think that the main selling point in the proposed scheme is to achieve deduplication checking. The efficiency of the scheme is also very important.  Thus, we add performance analysis in this manuscript.

Finally, while I appreciated the security analysis and the definition of the underlying security models the paper has many typos and grammatical mistakes. Therefore, the authors need to properly revisit the manuscript (on top of addressing the previous comments).

Finally, while I appreciated the security analysis and the definition of the underlying security models the paper has many typos and grammatical mistakes. Therefore, the authors need to properly revisit the manuscript (on top of addressing the previous comments).

Answer: thank you,  I revise the corresponding grammatical mistakes and typos.

Reviewer 2 Report

To protect the anonymity of receiver, the authors construct a privacy-preserving multi-receiver certificateless broadcast encryption scheme. The authors should further improve the writing performance. The construction and the security proof looks correct. The authors need to present the efficiency analysis for the construction w.r.t. computational and communication cost for the system. It is also better for the authors to compare their scheme with other existing ones in terms of security, efficiency and properties.

Author Response

Thank your  suggestion.

In the revised vision,  we give the corresponding computational cost of the main algorithm by comparing with Hung et al.scheme [55] and Islam et al. scheme [31].

Round 2

Reviewer 1 Report

Authors successfully addressed all of the previous comments. In addition to that, they added an important section where they evaluated the performance of the proposed scheme. 

Based on the current improvements, I suggest to accept the paper for publication.